# Clinical and Genetic Characteristics of COL2A1-Associated Skeletal Dysplasias in 60 Russian Patients: Part I

**DOI:** 10.3390/genes13010137

**Published:** 2022-01-13

**Authors:** Tatyana Markova, Vladimir Kenis, Evgeniy Melchenko, Darya Osipova, Tatyana Nagornova, Anna Orlova, Ekaterina Zakharova, Elena Dadali, Sergey Kutsev

**Affiliations:** 1Research Centre for Medical Genetics, Moskvorechye st., 1, 115522 Moscow, Russia; osipova.dasha2013@yandex.ru (D.O.); t.korotkaya90@gmail.com (T.N.); annaf-orlova@yandex.ru (A.O.); doctor.zakharova@gmail.com (E.Z.); genclinic@yandex.ru (E.D.); kutsev@mail.ru (S.K.); 2H. Turner National Medical Research Center for Сhildren’s Orthopedics and Trauma Surgery, Parkovaya 64–68, 196603 Saint-Petersburg, Russia; kenis@mail.ru (V.K.); emelcheko@gmail.com (E.M.)

**Keywords:** collagenopathy type II, *COL2A1* gene, exome sequencing, novel variants, skeletal dysplasia

## Abstract

The significant variability in the clinical manifestations of COL2A1-associated skeletal dysplasias makes it necessary to conduct a clinical and genetic analysis of individual nosological variants, which will contribute to improving our understanding of the pathogenetic mechanisms and prognosis. We presented the clinical and genetic characteristics of 60 Russian pediatric patients with type II collagenopathies caused by previously described and newly identified variants in the *COL2A1* gene. Diagnosis confirmation was carried out by new generation sequencing of the target panel with subsequent validation of the identified variants using automated Sanger sequencing. It has been shown that clinical forms of spondyloepiphyseal dysplasias predominate in childhood, both with more severe clinical manifestations (58%) and with unusual phenotypes of mild forms with normal growth (25%). However, Stickler syndrome, type I was less common (17%). In the *COL2A1* gene, 28 novel variants were identified, and a total of 63% of the variants were found in the triple helix region resulted in glycine substitution in Gly-XY repeats, which were identified in patients with clinical manifestations of congenital spondyloepiphyseal dysplasia with varying severity, and were not found in Stickler syndrome, type I and Kniest dysplasia. In the C-propeptide region, five novel variants leading to the development of unusual phenotypes of spondyloepiphyseal dysplasia have been identified.

## 1. Introduction

COL2A1-associated skeletal dysplasias are a large group of genetically heterogeneous diseases with an autosomal dominant type of inheritance caused by pathogenic variants in the *COL2A1* gene (OMIM:120140) [1]. The main clinical feature of this group is dwarfism, characterized by a shortening of the trunk [2]. Some patients have extra-skeletal symptoms, including myopia, vitreous abnormalities, retinal detachment, cataracts, glaucoma, sensorineural hearing loss, as well as orofacial anomalies-midface hypoplasia, micrognathia, cleft palate, or Pierre Robin sequence [3,4,5,6]. The existence of significant variability in the clinical manifestations of this group from perinatal lethal and severe nosological forms to mild clinical manifestations is shown. In addition, patients with overlapping phenotypes have been described. Thus, the clinical manifestations of this area of diseases represent a single “clinical continuum”, united in the group of “type II collagenopathies”, indicated in the international nosology and classification of genetic skeletal disorders [7]. Manifestations of classic congenital spondyloepiphyseal dysplasia (SEDC) can range from severe to mild, characterized by osteoarthritis in adolescence and moderate growth retardation. Spondyloepiphyseal dysplasia (SED) manifestations can be combined with marked changes in metaphysis that characterize spondyloepimetaphyseal dysplasia (SEMD) [8]. Mild forms of type II collagenopathies can be characterized by dysplasia of the proximal femoral epiphyses, which occurs in Legg–Calvé–Perthes disease, and aseptic necrosis of the femoral head [9,10]. The leading clinical symptom in patients with Stickler syndrome, type I (STL1) is congenital myopia, while the manifestations of SED are moderate [11,12]. At the same time, a typical clinical presentation of SEDC and Kniest dysplasia (KD) manifests from birth with a severe course, significant short stature, and a risk of atlanto-axial instability due to odontoid hypoplasia of the second cervical vertebra [13]. In addition, type II collagenopathies can be divided into two main groups based on typical radiographic patterns: delayed ossification of the juxtatruncal bones, including pear-shaped vertebrae, in the group of «SED and similar phenotypes», and abnormal growth of tubular bones with their dumbbell-like expansion in the group of «KD/STL1» [8].

The *COL2A1* gene is located on chromosome 12q13.11. It consists of 54 exons and encodes the α−1 chain of type II procollagen, formed in a helical configuration of three identical α−1 chains into a homotrimer after cleavage of the N- and C-terminal propeptides required for chains association and initiation of the triple helix formation, which are crosslinked with fibers similar to collagen fibrils [14,15]. Mature type II collagen molecules are synthesized by proliferating chondrocytes, providing both the structural function of the cartilage matrix and regulatory function through participation in the interaction and signal transmission in BMP-SMAD1 pathways that affect differentiation of chondrocytes [16].

To date, more than 600 pathogenic variants have been described in the *COL2A1* gene; most of them are missense variants localized in the triple helix domain of the protein consisting of 330 repeats of the Gly-XY amino acids [6,15,17]. There are two main molecular mechanisms underlying the pathogenesis of type II collagenopathies [6,18]. Missense variants that lead to the replacement of the Glycine residue in the Gly-XY triplet have a dominant-negative effect, leading to disruption of the helical structure and function of type II collagen, which are observed mainly in SED [19]. Pathogenic variants leading to premature termination of protein synthesis cause haploinsufficiency through the nonsense-mediated mRNA decay. This mechanism of the disease development is more commonly seen in patients with milder forms of collagenopathies, such as STL1 [20,21]. The KD development is more often associated with exon skipping variants [22].

To date, no clear genotype–phenotype correlations have been obtained in type II collagenopathies, which is most likely associated with the existence of the inter- and intrafamilial variability of clinical manifestations, differences in the specificity of clinical manifestations in patients with the same pathogenic variants, age-related evolution of phenotypes, and evidence of marked genetic heterogeneity due to some cases with an autosomal recessive type of inheritance [13,15,23,24,25,26]. Nevertheless, conducting a clinical and genetic analysis of various nosological variants of the skeletal collagenopathies spectrum can improve our understanding of the pathogenetic mechanisms and more accurately predict their course at the early stages, improving an individual’s medical care and quality of life [27].

## 2. Materials and Methods

A comprehensive examination of 60 Russian pediatric patients from unrelated families aged from 1 month to 17 years with phenotypic signs of type II collagenopathy was carried out. To clarify the diagnosis, following methods were used: genealogical analysis, clinical examination, neurological examination according to the standard technique with an assessment of the psychoemotional sphere, radiography, and targeted panel sequencing consisting of 166 genes responsible for the development of hereditary skeletal pathology.

Isolation of genomic DNA was carried out from whole blood using the DNAEasy (QiaGen, Hilden, Germany) according to the manufacturer’s standard protocol. The concentration of DNA and DNA libraries was measured on a qubit 2.0 instrument using reagents (qubit BR, qubit HS) from the manufacturer according to the standard protocol. For sample preparation, a technique based on multiplex polymerase chain reaction of target DNA regions was used. New generation sequencing was carried out on an Ion Torrent S5 sequencer with an average coverage of at least 80x; the number of targeted areas had coverage ≥90–94%. To annotate the identified variants, nomenclature presented on site http://varnomen.hgvs.org/recommendations/DNA version 2.15.11 was used. Sequencing data were processed using a standard automated algorithm from Ion Torrent.

To assess the population frequencies of identified variants, samples of the «1000 Genome» projects, ESP6500, and The Genome Aggregation Database v2.1.1 were used. To assess clinical significance of the identified variants, OMIM database and the HGMD^®^ Professional pathogenic variants database version 2021.3 were used. Assessment of the pathogenicity and causality of genetic variants was carried out in accordance with international recommendations for the interpretation of data obtained by massive parallel sequencing [28].

Validation of the identified variants in probands and genotyping of siblings and parents were carried out by automated Sanger sequencing according to the manufacturer’s protocol on the ABIPrism 3500xl device (Applied Biosystems, Waltham). The primer sequences were selected according to the reference sequence of the *COL2A1* gene target regions (NM_001844.5).

## 3. Results

We observed 60 unrelated probands (22 males and 38 females) with clinical and radiological signs of type II collagenopathy. More than half of the probands—58% (35/60)—had significant short stature with clinical and radiological signs of SEDC (*n* = 20), SEMD (*n* = 6), KD (*n* = 6), or intermediate forms of SEDC/KD (*n* = 3). The other 25 patients corresponded to milder forms of SED (*n* = 12), spondyloperipheral dysplasia (SPPD) (*n* = 2), Czech dysplasia (CD) (*n* = 1), and STL1 (*n* = 10). In 80% of the cases, the patients were the only one affected in the family, and, in 20% of the cases, there was segregation of the disease in two generations.

A typical clinical sign in all the patients was short stature. The degree of growth retardation varied significantly with different clinical forms and severity of the disease. The results are summarized in Figure 1.

The median growth in the SEDC, SEMD, and intermediate form SEDC/KD group of patients was: −5.29 SD, in KD group: −3.65 SD, in mild forms group of SED, SPPD, and CD: −1.38 SD, and in the STL1 group: 0.6 SD. The obtained results show that the early onset of severe forms of the collagenopathy type II spectrum leads to lower growth (−5.29 SD) in contrast to the mild late forms of SED and STL1, which are often accompanied by normal growth. A radiological data analysis of patients from 1 month to 17 years old revealed growth-related transformation of the abnormal ossification of the vertebrae and epiphyses of tubular bones, remarkable metaphyseal involvement with «corner fractures», «spotted» or enchondroma-like changes in six patients (10%), platyspondyly in thirty-nine (65%), and coxa vara in thirty-one (52%). The obtained data are presented in (Appendix A, Appendix A).

Additional extra-skeletal signs of type II collagenopathy were diagnosed in 55% (33/60) of the probands; among them, moderate to high myopia was found in 45% (27/60), bilateral sensorineural hearing loss in 22% (13/60), and cleft palate or Pierre Robin sequence in 22% (13/60) (Appendix A, Appendix A). Moreover, in the group of SEDC and SEMD patients, these signs were less common: congenital myopia was found in 30% of the cases, cleft palate in 19%, and sensorineural hearing loss was found only in 7%. By contrast, in the group of KD, intermediate phenotypes of SEDC/KD and STL1 (*n* = 19 probands) more often were detected: congenital myopia in 95% of the cases, cleft palate in 42%, sensorineural hearing loss in 58%. However, there were no extra-skeletal signs in patients with mild variants of SED, as well as SPPD and CD (*n* = 15).

As a result of the molecular genetic analysis, 54 variants in the *COL2A1* gene were identified, including 28 (52%) novel variants that were not described in the HGMD and GnomAD databases [17] (Appendix A, Appendix A). The validation and segregation of the detected variants in families by the automated Sanger sequencing confirmed their de novo status in 48 probands and autosomal dominant inheritance in 12 affected families. All the variants were allocated throughout the gene except for the N-propeptide region where no variants were found. Five variants were observed in more than one proband: c.1510G>A (p.Gly504Ser), c.1636G>A (p.Gly546Ser), c.3464G>T (p.Gly1155Val), c.2710C>T (p.Arg904Cys) were found in two probands each, and a novel previously undescribed variant c.2671G>A (p.Gly891Ser) was found in three probands. Moreover, 10 splice site variants were identified; two of them were found in the C-propeptide region. In addition, four deletions with or without frameshift, one variant with stop codon formation, and one duplication leading to frameshift were found (Figure 2).

Most of the variants led to amino acid substitutions in the triple helix domain—90% (54/60)—while glycine substitution in the Gly-XY repeats predominate in 63% (34/54). Among amino acid substitutions in the triple helix domain, the most frequent substitutions were glycine to serine 26% (14/54). Among the non-glycine substitutions, a group of four probands with arginine to cysteine substitutions was found, including c.823C>T (p.Arg275Cys) in patients with CD. In the C-propeptide region, six variants were found; five of them were detected for the first time. The spectrum of amino acid substitutions identified in the sample is presented in Table 1.

The distribution and analysis of the different types of variants in the *COL2A1* gene found in the sample revealed evident correlations between the variant types and typical phenotypic manifestations (Figure 3).

The phenotypes of patients with the most common glycine substitutions varied from classical SEDC to mild forms of SED. In two probands with a similar course of SEDC, the glycine to serine substitution was found at a hotspot: c.1510G>A (p.Gly504Ser) [15]. In three other probands with severe SEDC, a previously undescribed substitution of glycine to serine was found: c.2671G>A (p.Gly891Ser). In addition, there was a polymorphism of clinical manifestations in two probands with a similar variant, where a substitution of glycine to serine at the 546 nucleotide position was detected. One of the patients had typical signs of SEMD and another one had a mild phenotype of SED with normal growth, as confirmed by the data of Chen J. et al. [27], where an interfamily clinical polymorphism was identified in patients with this pathogenic variant. In two patients with newly identified substitutions leading to glycine substitution: c.1348G>C (p.Gly450Arg) and c.3554G>A (p.Gly1185Glu), radiography revealed the presence of enchondroma-like changes in metaphyses, anisospondylia, limb length discrepancy, and abnormal ossification pattern of the pubic bones which are typical for dyspondyloenchondromatosis (Figure 4).

## 4. Discussion

Type II collagenopathies caused by pathogenic variants in the *COL2A1* gene are characterized by a wide range of clinical and radiological signs, and their modification can occur across ages [13]. An analysis of 60 probands showed that, in childhood, forms with more severe clinical manifestations are more common: SEDC, SEMD, and KD (58%), or unusual phenotypes of mild forms of SED (25%), often hidden under the guise of rheumatoid-like arthritis or degenerative diseases of the hip joints. STL1 was less prevalent (17%). The average age at STL1 diagnosis according to Zhang et al. [15] is 20.7 years. Our data confirm that, in the diagnosis of COL2A1-associated skeletal displasias, it is important to consider the existence of both: the classical phenotypes of SEDC and KD, which appear from birth and gradually progress, and rare forms with a predominant lesion of the hip joints and normal growth. A typical sign of the phenotype’s severity in the type II collagenopathies spectrum is a short stature, which was 5.29 SD in the group of patients with SEDC, KD, and the intermediate SEDC/KD form. In addition, moderate to high congenital myopia had become a useful diagnostic feature among all the phenotypes and was found in 45% of the cases, which is especially important for diagnosing STL1 due to the milder skeletal phenotype. Our data confirm the results of other studies that showed a higher frequency of the main extra-skeletal manifestations: high-grade congenital myopia, cleft palate, and sensorineural hearing loss, in patients with KD and STL1 [13,25].

Mild forms of SED were diagnosed in 12 probands; their growth was within the middle-lower limit or was slightly reduced (from 0.17 to 2.5 SD). Moreover, they showed no extra-skeletal signs. Despite the disproportionate shortening of the trunk, this group had a greater involvement of the proximal femoral epiphyses and often had unusual phenotypes that complicate diagnosis.

In the group of patients with various clinical variants of type II collagenopathies, 54 variants in the *COL2A1* gene were identified; 28 of them were not previously described in the HGMD database [17]. Clinical and genetic analyses in separate groups of patients showed that 63% of the variants led to the substitution of glycine in Gly-XY repeats. The presence of this type of substitution in patients has been associated with the development of SEDC and SEMDC with varying severity. However, they were not found in STL1 and KD patients. Interestingly, some variants leading to glycine to other amino acid substitutions have been associated with rare phenotypes. For example, a substitution of glycine to serine in position 945 was found in an 11-year-old boy with osteochondritis dissecans of the left femoral head; a substitution of glycine to valine in position 348 was found in a 7-year-old proband with Perthes-like changes in the hip joints, and a substitution of glycine to valine in position 207 in a 5-year-old boy with arthralgias and developing joint contractures. The variant c.620G>T (p.Gly207Val) was identified for the first time; however, other amino acid substitutions with similar phenotypes were described in the same codon. Jurgens J. et al. [30] in 2015 described three patients with progressive pseudo-rheumatoid dysplasia and SED, Stanescu type, with c.619G>A (p.Gly207Arg) variant. Rolvien T. et al. [31] observed a mother and son of 8-year-old with SED with early osteoarthritis, where a variant c.620G>A (p.Gly207Glu) was found. Travessa A.M. et al. [32] reported on an 8-year-old boy with SED, Stanescu type characterized by progressive contractures and premature degenerative joint disease with normal growth.

In the SEDC–SEMD group, only one de novo missense variant, which led to a non-glycine substitution at the Y position of the Gly-XY triplet: p.2974A>G (p.Arg992Gly), was identified. This variant has been previously described by Sulko J. et al. [33] in 2005 in monozygotic twin girls from Poland. The clinical manifestations of the Polish patients and our 4-year-old proband were similar and characterized by marked dwarfism and severe manifestations of SEMDC. In addition, our patient had a high degree of congenital myopia, which was not reported in previously described patients [33]. The identified variant was not registered in the GnomAD database. Algorithms for predicting pathogenicity define this variant as pathogenic (SIFT, PolyPhen2, FATHMM). According to the pathogenicity predictors SpliceAl, Max Entropy Scan, and SPIP, the detected c.2974A>G (p.Arg992Gly) variant does not affect the exon splicing sites. Thus, summarizing data allows us to categorize the identified variant into the SEDC–SEMD group.

Among the “non-glycine” substitutions, arginine to cysteine substitution in both the X and Y positions in the Gly-XY triplet were more common, that according to Hoornaert K.P. et al. [34] affects the final phenotype: without eyes involvement in the Y-position, with eyes involvement in the X-position [30]. Interestingly, the substitution of arginine to cysteine in the Y-position: c.3397C>T (p.Arg1133Cys) was found in a 9-year-old boy with arthralgia and signs of SED tarda, who had previously been excluded from the diagnostic search for lysosomal storage disease. In the diagnostic process, it was established that the amino acid substitution identified in the child was inherited from his father, who underwent hip arthroplasty at the age of 35 due to progressive coxarthrosis. Girisha K. M. et al. [26] presented an exceptionally rare case of two affected siblings with this variant in a homozygous with moderate signs of SED, and, like in our patient, a mild form of Morquio A syndrome was first considered as a potential diagnosis. A substitution of arginine to cysteine in the Y-position was found in a patient with CD: c.823C>T (p.Arg275Cys), as in previously described cases with CD, which allows us to consider it as a hotspot for CD [15]. In addition, in two probands from our sample, the substitution of arginine to cysteine in the X-position of the triplet was detected: c.2710C>T (p.Arg904Cys) in a hotspot, which was repeatedly reported by other authors (11 cases). Moreover, in one of the probands, apart from brachydactyly, brachymetatarsia of the fourth ray was noted (type E brachydactyly on the feet) [15]. In three probands, previously undescribed missense variants in the C-propeptide region were revealed: c.3897G> T (p.Trp1299Cys), c.3950T>G (p.Met1317Arg), c.4133T>A (p.Leu1378Gln); each of them was accompanied by an atypical phenotype: a familial form of early osteoarthritis, signs of significant fragmentation of the proximal femoral epiphyses, and aseptic necrosis of the femoral heads with total epiphyseal lesions. Of particular interest were two cases of STL1 with an unusually tall height in boys: 190 cm, caused by newly identified variants in the splice site, localized in the C-propeptide. The first 14-year-old boy has a variant: c.4317+1G>T, accompanied by a severe course with early osteoarthritis, retinal detachment, and cataract of the right eye, which led to ophthalmic surgery and endoprosthetics of the right femur caput. A second 12-year-old boy with a high degree of congenital myopia, flattened middle third of the face in early childhood, first degree bilateral sensorineural hearing loss, and marfanoid phenotype has a variant: c.4074+1G>A. In 2010, researchers Hoornaert K.P. et. al. [21] described a pathogenic variant in the same codon: c.4074+1G>T, leading to the development of the STL1 phenotype. In addition, in the proband with the c.4074 + 1G>A variant in the *COL2A1* gene, another de novo stop gained variant was also identified in exon 41 of the FBN1 gene: c.5060_5061delGCinsAA (p.Cys1687Ter), which was not described in the HGMD database [17]. However, another variant, c.5061C>A (p.Cys1687Ter), leading to Marfan syndrome development (CM117725), was described in this codon, which, in total, does not exclude the presence of “double trouble” pathology in the child [17,21]. Three cases of STL1 in our sample were caused by variants with the loss of protein function: nonsense, duplication, and frameshift deletion. Two probands had autosomal dominant inheritance of STL1 syndrome from one of their parents with detected duplication leading to frameshift: c.2813dupC (p.Gly939Trpfs*) and nonsense variant: c.2839C>T (p.Gln947Ter). Moreover, at birth, children had a Pierre Robin sequence, and their affected parents did not have a cleft palate, but recurrences of retinal detachment were manifested. In these cases, a family history of intrafamilial clinical polymorphism can be a good indicator for searching variants in the *COL2A1* gene. The small number of patients with STL1 in our sample makes it difficult to determine clinical and genetic correlations and allows us to identify only some trends that are consistent with the results of other studies. Thus, in four out of the ten patients with STL1, caused by splicing site variants, as well as in Korean patients with this type of nucleotide substitution, early retinal detachment was noted, which was detected already at the beginning of the second decade of life, which is a rather serious complication of the disease [35].

KD among our probands was mainly caused by splice site variants in the triple helix domain; two of them were detected for the first time: c.1266+5G>C and c.970-8T>G. In one case, a newly identified deletion without frameshift was found in exon 23 of the *COL2A1* gene c.1421_1426del (p.Gly474_Pro475del). In the control sample of the GnomAD database, these variants have never been encountered. According to pathogenicity predictors (SpliceAl, MMSplice, Human Splicing Finder), variants c.1266+5G>C and c.970-8T>G affect the splice donor site, which probably leads to exon skipping.

In addition, in three cases, intermediate phenotypes between SEDC and KD were observed. In one case, an amino acid substitution c.905C>T (p.Ala302Val) was found. For this substitution, a functional analysis was previously performed by Chen L et al. [29], showing its effect on the splicing process with the skipping of 21 nucleotides. In the second case, previously described by Winterpacht A. et al. [36], a deletion without frameshift in a 5-year-old boy with KD was identified: c.3627_3644del (p.Pro1211_Pro1216del), and, in the third case, a novel deletion without frameshift was found: c.3442_3444delTCT (p.Ser1148del). Despite the severe phenotype of these three cases and the presence of extra-skeletal signs, the X-ray “pattern” resembled the SEDC phenotype. We diagnosed SPPD in two probands with newly identified variants in the *COL2A1* gene: the substitution of glycine to cysteine c.1090G>T (p.Gly364Cys) as a familial case and splice site variant: c.817-1G>A as a de novo variant.

Indeed, for the identified splicing site variants, it is necessary to perform functional studies, including the creation of appropriate mini-gene constructs, to study the molecular effect and assess the clinical significance of the variants. Ex vivo mini-gene analysis shows whether a splice site variant causes a frameshift or exon skipping in the *COL2A1* gene, which may be important in determining the phenotype of COL2A1-skeletal dysplasia [37,38]. Typically, small deletions in the reading frame lead to the development of KD. Variants leading to a premature stop codon in the *COL2A1* gene are mainly associated with the mildest form of STL1 [6,15]. Currently, we are testing the pathogenicity of all the novel variants from our sample.

## 5. Conclusions

Based on the clinical and genetic characteristics of 60 Russian pediatric patients with variants in the *COL2A1* gene, an etiological cause and a range of typical and atypical phenotypes have been established, which are important in the formation of approaches to the type II collagenopathies diagnosis. It was shown that clinical forms of SED predominate in childhood, both with more severe clinical manifestations: SEDC, SEMD, and KD (58%), and with unusual mild forms of SED phenotypes with normal growth (25%), accompanied by early osteoarthritis with a rheumatoid-like course or hiding under the guise of Legg–Calvé–Perthes disease. However, STL1 was less often detected (17%) and manifested mainly by ocular pathology. Radiographic signs of the classical forms of SEDC, SEMD, and KD were transformed over time as the skeleton matured. The main extra-skeletal signs were diagnosed in 55% of the probands; the more significant of them were moderate to high myopia, which was in 45% of the probands; cleft palate or Pierre Robin sequence were less common: 22%, as was bilateral sensorineural hearing loss: 22%. These signs were absent at the time of examination in patients with mild forms of SED. The importance of family history has been established due to the intrafamilial polymorphism of clinical features, which is also a key point for the variants searching in the *COL2A1* gene. Next-generation exome sequencing provides a cost-effective and rapid approach to identifying pathogenic variants in the *COL2A1* gene responsible for type II collagenopathies, which is important for genetic counseling and can reduce the risk of recurrent cases in families and improve the clinical follow-up of patients. As a result of molecular genetic testing, 28 novel variants in the *COL2A1* gene were identified, which make up 52% of the Russian sample. No characteristic regional types of genotypes were established for Russian patients due to high genetic heterogeneity, with the exception of a previously undescribed variant: c.2671G>A (p.Gly891Ser), identified in three unrelated Russian patients. However, we have expanded our understanding of the ratio and occurrence of the COL2A1-associated skeletal dysplasias phenotypes in childhood, including atypical and rare forms, such as dyspondyloenchondromatosis. Moreover, a potential genotype–phenotype correlation has been identified, which can improve our understanding of the type II collagenopathies pathogenetic mechanisms and prognosis. The most prevalent variants were found in 63% of the patients and resulted in glycine substitutions in the triple helix domain. Among the “non-glycine” substitutions in the triple helix domain, the substitutions of arginine to cysteine were more prevalent and caused unusual phenotypes with or without eyes involvement. In two probands with STL1, a substitution in the X-position of the Gly-XY triplet was found: c.2710C>T (p.Arg904Cys); in a proband with CD, a substitution in the Y-position was found: c.823C>T (p.Arg275Cys), and, in a proband with SED tarda, similar to the mild form of Morquio A syndrome described earlier by Girisha K.M. et al., [26] variant c.3397C>T (p.Arg1133Cys) was found. In the C-propeptide region, five novel variants leading to the unusual SED phenotypes development have been identified. In patients with KD and STL1 from our sample, splice site variants were predominantly found; for these variants, a functional analysis will be presented in our subsequent study. Further data accumulation in the phenotypic variability of type II collagenopathies caused by already known and newly identified variants in the *COL2A1* gene will contribute to a more detailed description of genotype–phenotype correlations, which will allow us to predict the severity of the disease in patients with various pathogenic variants and to determine the tactics of patient management.

## Figures and Tables

**Figure 1 genes-13-00137-f001:**
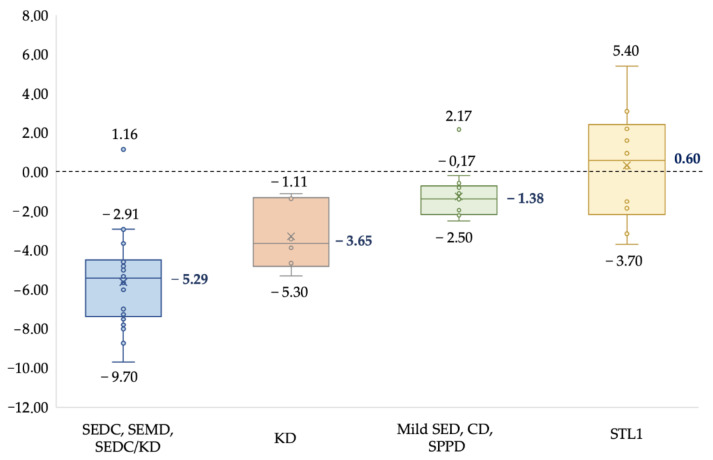
Deviation of growth parameters (SDS) from the average standard values in patients with various phenotypes of type II collagenopathies.

**Figure 2 genes-13-00137-f002:**
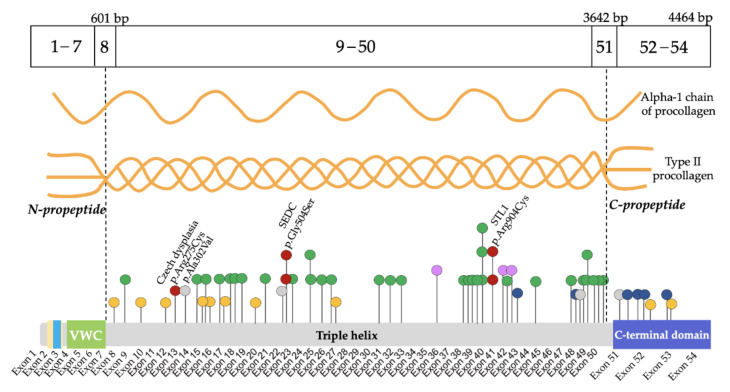
Variants localization in exons of the *COL2A1* gene and distribution of amino acid substitutions in protein domains. Variant c.905C>T (p.Ala302Val) has been defined into a gray group since it leads to a non-frameshift deletion [29].

**Figure 3 genes-13-00137-f003:**
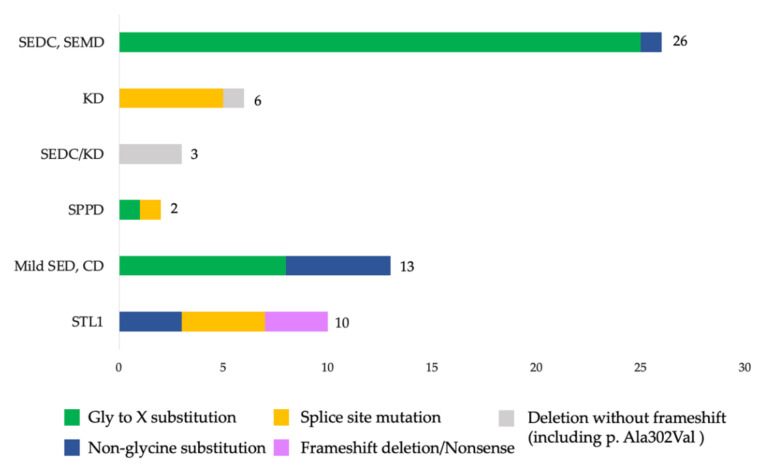
Distribution of various variants in patients with different clinical forms of type II collagenopathies.

**Figure 4 genes-13-00137-f004:**
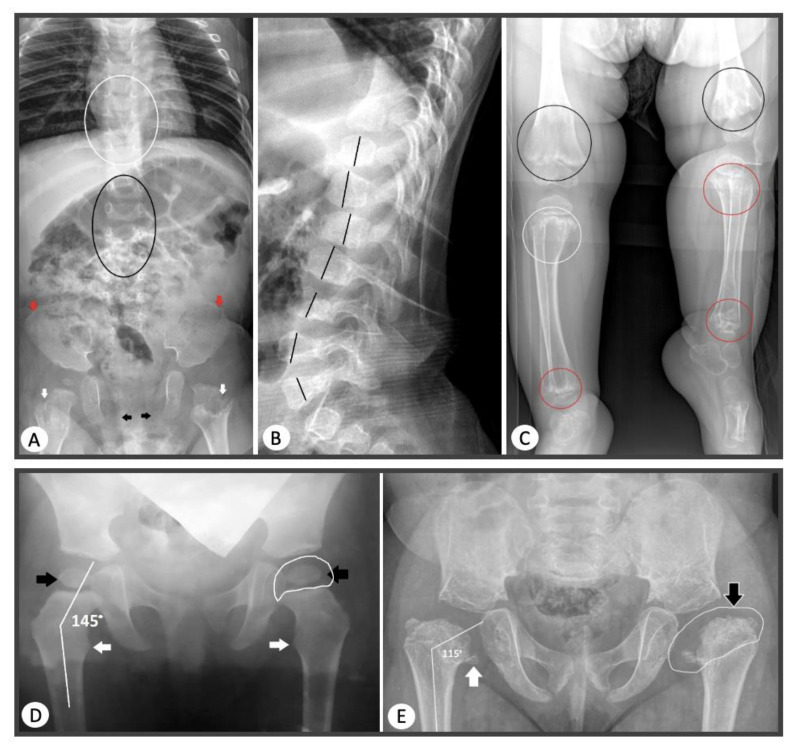
Radiological signs of COL2A1-skeletal dysplasia in probands from our sample. (**A**) AP radiograph of thoracolumbar spine, pelvis, and hips of the patient with dysspondyloenchondromatosis. Anisospondyly and mild wedge-shape vertebral bodies of the thoracic (white circle) and lumbar (black circle) spine; lacy iliac crests (red arrows); deficient ossification of the pubic bones (black arrows); chondromatous lesions of the femoral necks (white arrows). (**B**) Lateral radiograph of thoracolumbar spine of the patient with dysspondyloenchondromatosis. Anisospondyly of the thoracic and lumbar spine (the heights of the vertebral bodies marked with the black lines). (**C**) Radiograph of both legs of the patient with dysspondyloenchondromatosis. Extensive chondromatous lesions of the metaphyses of the distal femora (black circles), proximal (white circles), and distal (red circles) tibiae and fibulae; remarkable length discrepancy on the lower legs. (**D**) AP radiograph of the hips of the patient with mild phenotype. Coxa valga (neck-shaft angle 145°, normal length of the femoral necks), normal ossification of the femoral head and neck (black arrows), and lesser trochanter (white arrows); normal shape of the femoral head (white line depicts presumed contour of the cartilaginous femoral head). (**E**) AP radiograph of the hips of the patient with severe phenotype. Hypoplastic iliac wings, coxa vara et breva (neck-shaft angle 115°, short femoral neck), abnormal ossification of the femoral head and neck (black arrow), and lesser trochanter (white arrow); abnormal shape of the femoral head-coxa plana (white line depicts presumed contour of the cartilaginous femoral head).

**Table 1 genes-13-00137-t001:** Prevalence of different types of variants in the probands.

Variant Type	Frequency
Triple Helix Domain	
Gly to Ser	14
Gly to Val	7
Gly to Arg	6
Gly to Glu	3
Gly to Cys	3
Gly to Asp	1
Arg to Cys	4
Arg to Gly	1
Ala to Val	1
Splice	8
In-frame deletion	3
Out-of-frame deletions	1
Out-of-frame duplications	1
Nonsense	1
С-propeptide	
Missense	4
Splice	2
Total	60

## Data Availability

Not applicable.

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
