# Peer review of "Clinical and Genetic Characteristics of COL2A1-Associated Skeletal Dysplasias in 60 Russian Patients: Part I"

_genes, 2022, doi:10.3390/genes13010137_

Round 1

Reviewer 1 Report

This is a very nice, detailed study of characteristics of COL2A1-associated skeletal dysplasias in Russian. This study is extremely useful in explaining the genotype-phenotype correlations for COL2A1-associated skeletal dysplasias.

My only main recommendation is to description of the characteristic mutations and phenotypes in Russian patients. It is also important to mention that their no regional variation if Russia patients does not have the characteristic phenotypes and genotypes.

Author Response

Response to Reviewer 1:

The authors are grateful to the reviewer for carefully reading the work and making comments. We believe that considering the comments of the reviewer made it possible to significantly improve quality of our manuscript. According to the recommendations of the reviewer, we have summarized the main comments and made the following changes in the "Conclusion" section of the manuscript.

Point 1: My only main recommendation is to description of the characteristic mutations and phenotypes in Russian patients. It is also important to mention that their no regional variation if Russia patients does not have the characteristic phenotypes and genotypes.

 Response 1: Indeed, no characteristic regional genotypes were established for Russian patients due to high genetic heterogeneity, with the exception of a previously undescribed variant: c.2671G>A (p.Gly891Ser), identified in three unrelated Russian patients. However, we have expanded our understanding of the ratio and occurrence of the COL2A1-associated skeletal dysplasias phenotypes in childhood, including atypical and rare forms, such as dyspondyloenchondromatosis.

We are grateful to the reviewer for positive review and helpful comments.

Sincerely, collective of authors

Reviewer 2 Report

In this study, Markova et al. analysed 60 patients with type II collagenopathy to detect pathogenic variants in the COL2A1 gene. The data collected were used to contribute to a genotype-phenotype correlation.

Although the manuscript is quite understandable, some data need to be presented more clearly.

I have the following comments and suggestions to propose.

  • Did the authors analyse 60 patients with a panel of 166 genes, and did they find variants just in COL2A1? I guess not. Maybe the 60 patients are from more complex and numerous clinical cases or something else. Please explain better in the text or discuss this issue.
  • The word “variant” should replace the term “mutation” throughout the manuscript as much as possible. In particular, in some phrases, the word “mutation” was synonymous with nucleotide change but as a pathogenic variant in other sentences.
  • In figures 2 and 3, colours that underline the type of COL2A1 variants are different. I prefer the distribution used in figure 3 because it is important to distinguish Glycine from non-glycine substitutions and frameshift deletions from non-frameshift deletions.
  • The interpretation of novel splice variants (c.1266+5G> C and c.970-8T> G) is unclear without a functional study. Therefore, the authors should use at least a bioinformatic tool like HSF (Human splicing finder) to evaluate the possible pathogenicity of the variants.
  • It is important to better discuss the findings on the splice site variants. Some of them are in the KD group, but others are in the STL1 group. This apparent discordance is due to the possibility that a splice site variant can generate a frameshift or a deletion without frameshift. The first consequence, usually, is associated with a milder phenotype. Thus, the relevance of functional studies to detect protein consequences with some splice variants should be underlined in the discussion section. In the discussion, it would be better to add these references (PMID: 34573377, PMID: 32196734) where COL2A1 splice variants were studied by minigene assay, obtaining different results in patients with very different phenotypes.
  • The authors should compare their findings of STL1 group with those published in the following studies (PMID: 34680973, PMID: 32756486) to strengthen their conclusions.
  • Since the c.905C> T (p.Ala302Val) is a splice site variant leading to a non-frameshift deletion (reference 33, cited by the authors), it would be better to include this variant in the “grey” group (figure 3), adding a note.
  • In the SEDC-SEMD group (figure 3), there is a non-glycine substitution. Is it possible this variant is a splice site variant as p.Ala302Val? Is it possible for the patient an alternative diagnosis? Please, perform bioinformatic analysis of the variant and discuss.
  • The authors should state that the novel pathogenic variants detected are absent in a database as GnomAD.

Author Response

Response to Reviewer 2 :

We thank the reviewer for a detailed analysis of our manuscript. Your comments and suggestions helped us significantly improve the quality of the obtained data analysis and their presentation. In the revised version of this paper, we tried, as much as possible, to consider the comments and suggestions presented in the review.

Point 1: Did the authors analyse 60 patients with a panel of 166 genes, and did they find variants just in COL2A1? I guess not. Maybe the 60 patients are from more complex and numerous clinical cases or something else. Please explain better in the text or discuss this issue.

Response 1: The analysis of the genes target panel using the NGS method helped us in the diagnosis of causal variants in 47 genes in more than 200 Russian patients with skeletal dysplasias, but this study presents the results of the identified variants only in the COL2A1 gene in 60 patients, which expanded our understanding of the ratio and occurrence of the COL2A1-associated skeletal dysplasias in childhood, in this regard, we made additional changes in the "Conclusion" section.

Point 2: The word “variant” should replace the term “mutation” throughout the manuscript as much as possible. In particular, in some phrases, the word “mutation” was synonymous with nucleotide change but as a pathogenic variant in other sentences.

Response 2: We agree with the reviewer opinion regarding the generally accepted term "variant" and have tried to correct this in our manuscript.

Point 3: In figures 2 and 3, colours that underline the type of COL2A1 variants are different. I prefer the distribution used in figure 3 because it is important to distinguish Glycine from non-glycine substitutions and frameshift deletions from non-frameshift deletions.

Response 3: We thank for the helpful comment from the reviewer, we changed the colors in Figures 2 and 3 based on the reviewer's comments, that we believe will be extremely helpful in explaining genotype-phenotype correlations for COL2A1-associated skeletal dysplasias.

Point 4: The interpretation of novel splice variants (c.1266+5G> C and c.970-8T> G) is unclear without a functional study. Therefore, the authors should use at least a bioinformatic tool like HSF (Human splicing finder) to evaluate the possible pathogenicity of the variants.

Response 4: As recommended by the reviewer, we have made the following changes to the "Discussion" section:

In the control sample of the GnomAD database, these variants have never been encountered. According to pathogenicity predictors (SpliceAl, MMSplice, Human Splicing Finder), variants c.1266+5G>C and c.970-8T> G affect the splice donor site, which probably leads to exon skipping.

Point 5: It is important to better discuss the findings on the splice site variants. Some of them are in the KD group, but others are in the STL1 group. This apparent discordance is due to the possibility that a splice site variant can generate a frameshift or a deletion without frameshift. The first consequence, usually, is associated with a milder phenotype. Thus, the relevance of functional studies to detect protein consequences with some splice variants should be underlined in the discussion section. In the discussion, it would be better to add these references (PMID: 34573377, PMID: 32196734) where COL2A1 splice variants were studied by minigene assay, obtaining different results in patients with very different phenotypes.

Response 5: We agree with important recommendations of the reviewer regarding the relevance of the functional analysis, which we will do later, due to the large number of novel variants. At present, the final stages of the functional analysis of novel variants are being processed using mini-gene constructs for the discovered variants in the COL2A1 gene, that can potentially affect the splicing process. Soon, we plan to publish the second part of this manuscript, which will be dedicated to the description of the functional analysis of novel variants in the COL2A1 gene. As a result, we have changed the title of the revised version of this manuscript to Part I.

As recommended by the reviewer, we have made the following changes to the "Discussion" section:

Indeed, for the identified splicing site variants, it is necessary to perform functional studies, including the creation of appropriate mini-gene constructs to study the molecular effect and assess the clinical significancy of the variants. Ex vivo mini-gene analysis shows whether a splice site variant causes a frameshift or exon skipping in the COL2A1 gene, that may be important in determining the phenotype of COL2A1-skeletal dysplasia [Sun W, 2020; Bruni V, 2021]. Typically, small deletions in the reading frame lead to the development of KD. Variants leading to a premature stop codon in the COL2A1gene are mainly associated with the mildest form of STL1 [Zhang B, 2020; Barat-Houari M, 2016]. Currently, we are testing the pathogenicity of all novel variants from our sample.

Point 6: The authors should compare their findings of STL1 group with those published in the following studies (PMID: 34680973, PMID: 32756486) to strengthen their conclusions.

Response 6: We considered this recommendation of the reviewer and tried to make additional clarifications in the "Discussions" section:

The small number of patients with STL1 in our sample makes it difficult to determine clinical and genetic correlations and allows us to identify only some trends that are consistent with the results of other studies. Thus, in 4 out of 10 patients with STL1, caused by splicing site variants, as well as in Korean patients with this type of nucleotide substitutions, early retinal detachment was noted, which was detected already at the beginning of the second decade of life, that is a rather serious complication of the disease [Choi SI, 2021].

Point 7: Since the c.905C> T (p.Ala302Val) is a splice site variant leading to a non-frameshift deletion (reference 33, cited by the authors), it would be better to include this variant in the “grey” group (figure 3), adding a note.

Response 7: We thank the reviewer for the helpful comment, we changed the color in Figure 3 based on the reviewer's comments, fully agree that this will improve presentation of our data analysis.

Point 8: In the SEDC-SEMD group (figure 3), there is a non-glycine substitution. Is it possible this variant is a splice site variant as p.Ala302Val? Is it possible for the patient an alternative diagnosis? Please, perform bioinformatic analysis of the variant and discuss.

Response 8: We agree with the recommendation of the reviewer and made the appropriate changes in the "Discussions" section:

In the SEDC-SEMD group, only one de novo missense variant, which led to a non-glycine substitution at the Y position of the Gly-XY triplet: p.2974A> G (p.Arg992Gly) was identified. This variant has been previously described by Sulko J. et al. in 2005 in monozygotic twin girls from Poland. The clinical manifestations of the Polish patients and our 4-year-old proband were similar and characterized by marked dwarfism and severe manifestations of SEMDC. In addition, our patient had a high degree of congenital myopia, which was not reported in previously described patients [Sulko J. et al.,2005]. The identified variant was not registered in the GnomAD database. Algorithms for predicting pathogenicity define this variant as pathogenic (SIFT, PolyPhen2, FATHMM). According to the pathogenicity predictors SpliceAl, Max Entropy Scan, and SPIP, the detected c.2974A> G (p.Arg992Gly) variant does not affect the exon splicing sites. Thus, summarizing data allows us to categorize the identified variant into the SEDC-SEMD group.

Point 9: The authors should state that the novel pathogenic variants detected are absent in a database as GnomAD

Response 9: We considered this recommendation of the reviewer and tried to make additional clarifications in the manuscript:

"As a result of molecular genetic analysis, 54 variants in the COL2A1 gene were identified, including 28 (52%) novel variants that were not described in the HGMD and GnomAD databases".

We are very grateful to the reviewer for detailed analysis and valuable comments, which will help us to improve our manuscript.

Respectfully yours, collective of authors.

Round 2

Reviewer 2 Report

I think the authors made all possible efforts to improve the manuscript. I suggest, but it is just my opinion, to maintain the original title without adding part 1. The further study may be entitled: Functional studies on COL2A1 variants associated with Skeletal Dysplasias.